# Mpox Epidemics: A Call to Restore Humanity’s Lost Herd Immunity to Orthopoxviruses

**DOI:** 10.3390/v17091257

**Published:** 2025-09-18

**Authors:** Misaki Wayengera, Henry Kyobe-Bosa, Winters Muttamba, Olushayo Oluseun Olu, Abdou Salam Gueye, Nicaise Ndembi, Neema Kamara, Morenike Oluwatoyin Folayan, Bruce Kirenga, Sitong Luo, Qingyu Li, Chikwe Ihekweazu

**Affiliations:** 1Department of Immunology & Molecular Biology, College of Health Sciences, Makerere University, Kampala P.O. Box 7072, Uganda; 2Interdisciplinary Consortium for Epidemics Research, Kampala P.O. Box 189784, Uganda; brucekirenga@yahoo.co.uk; 3Incident Management Team, Ministry of Health, Kampala P.O. Box 7272, Uganda; 4Uganda Peoples Defense Forces, Kampala P.O. Box 3798, Uganda; 5Makerere University Lung Institute, Kampala P.O. Box 7749, Uganda; 6World Health Organization Regional Office for Africa, Brazzaville P.O. Box 06, Republic of Congocihekweazu@who.int (C.I.); 7International Vaccine Institute, Kigali P.O. Box 3413, Rwanda; nicaise.ndembi@ivi.int; 8Africa Centres for Disease Control and Prevention, Addis Ababa P.O. Box 3243, Ethiopia; kamaran@africacdc.org; 9Department of Child Dental Health, Awolowo University, Ile-Ife 220005, Nigeria; toyinukpong@yahoo.co.uk; 10Vanke School of Public Health, Tsinghua University, Beijing 100084, China; sitongluo@mail.tsinghua.edu.cn (S.L.); liqingyu21@mails.tsinghua.edu.cn (Q.L.)

**Keywords:** mpox immunity, waning immunity, smallpox vaccination

## Abstract

Global efforts to eradicate smallpox—an Orthopoxvirus infection—began in the mid-20th century, with the last naturally occurring case reported in 1977. This was achieved through global solidarity efforts that expanded the smallpox eradication vaccination program. Approximately 50 years following the cessation of mass smallpox vaccination and in the absence of access to a sustainable boosting program, the population immunologically naïve to Orthopoxviruses has increased significantly. With increasing global movements and travels, we argue that the emergence of two back-to-back yet distinct mpox epidemics in the 21st century is a sign of humanity’s lost herd immunity to Orthopoxviruses. This needs concerted efforts to restore.

## 1. Introduction

*Orthopoxviruses* are a group of zoonotic, phylogenetically related, double-stranded DNA viruses. The genus *Orthopoxvirus* belongs to the family *Poxviridae*, under which there are several genera, including *Parapoxvirus*, *Avipoxvirus*, *Capripoxvirus*, *Leporipoxvirus*, *Suipoxvirus*, *Molluscipoxvirus*, and *Yatapoxvirus*. Historically, the most widely reported infection caused by an *Orthopoxvirus* is smallpox—a disease of humans from time immemorial. The global eradication program for smallpox started in the mid-20th century, with the World Health Organisation (WHO) proposing a global smallpox eradication program in 1959 [1]. The timeline of events towards eradication is summarized in Figure 1.

Since 2022, humanity has faced two distinct but consecutive mpox (formerly monkeypox) outbreaks caused by the mpox virus (MPXV) [2,3]. The first outbreak (2022–2023) driven by Clade IIb spread, rapidly across Europe and North America. The current epidemic driven by Clade Ib began in August 2023 in South Kivu Province, Democratic Republic of Congo (DRC), and has since spread rapidly across East, Central, and Southern Africa [4,5,6].

These two discrete but closely related epidemics of mpox have not only increased interest in the role of *Orthopoxviruses* in causing pandemics but also serve as an indication of humanity’s prevailing risk to and/or immunity against *Orthopoxviruses* [7]. A growing school of thought attributes recent mpox outbreaks to waning *Orthopoxvirus* immunity, particularly among populations born after routine smallpox vaccination ceased [8,9]. It is estimated that up to 70% of the world’s population is no longer protected against smallpox and closely related *Orthopoxviruses* through cross-immunity [10].

Vaccination has emerged as a medical countermeasure for responding to mpox; however, this is affected by the high cost and unavailability of enough doses of the common vaccines like Bavarian Nordic’s MVA-BN vaccine that was initially licensed for smallpox. Evidence shows smallpox vaccine-induced immunity has the potential to protect against other *Orthopoxviridae* viruses, including mpox [11,12,13,14] (Table 1).

## 2. A Case for a Global Vaccination Program to Restore Humanity’s Herd Immunity Against *Orthopoxviruses*: Strategies for Restoring *Orthopoxviruses* Herd Immunity

In the 1960s, the initial goal of the smallpox vaccination program was to vaccinate 80% of the population to achieve herd immunity [19]. Given the dwindled herd immunity of most of the world’s population, we argue that the world needs to come together to see to it that this shield is restored rather than leaving the mandate to individual countries, some of which are too poor to afford the vaccines. Below we highlight the strategies for restoring *Orthopoxvirus* herd immunity.

## 3. Targeted Smallpox Vaccination Strategies

The ring vaccination strategy has been credited for the smallpox eradication and involves creating a buffer of immunity around a case by vaccinating the contacts and ultimately preventing disease spread. The current mpox epidemic gives the world an opportunity to deploy the same strategy by utilizing proven smallpox vaccines. An epidemiology-based approach that hinges on robust case-finding strategies to identify mpox cases and identification and vaccination of high-risk individuals could potentially build herd immunity in a global population that lacks immunity against *Orthopoxviruses*. Such a strategy should be supported by dose-sparing approaches given the low stockpiles of these vaccines to ensure that high vaccination coverage [20].

## 4. Strategies to Ensure Global Equity and Access, Particularly in Endemic and Low-Income Regions

In the short term, there is a need for a clear demonstration of global solidarity to respond to the mpox outbreak. Such solidarity should acknowledge the failures witnessed during the COVID-19 pandemic, where low-resourced countries lacked sufficient vaccines for their populations. A globally coordinated framework to ensure mpox vaccines are available to endemic and low-income countries should be developed. Implementation should be informed by epidemiological need rather than financial capacity. A shift from centralized structures to structures that promote regional coordination and context-driven responses should be supported. This has been demonstrated on the African continent, where the Africa Centers for Disease Control and Prevention (Africa CDC) and WHO Regional Office for Africa (WHO/AFRO) have coordinated the mpox continental preparedness and response to achieve localized and equitable health responses [21].

## 5. Development of Newer, Cheaper, Safer and More Effective Vaccines: Safer, More Effective Vaccines with Broader Cross-Protection

In the medium to long term, mpox vaccine research and development should be scaled up, particularly in Africa. Vaccine developers should consider expanding their manufacturing footprint in Africa, where more than 70% of health technology requirements are imported. Countries should allocate sufficient research and development funding through several mechanisms such as multilateral financing schemes, assurance of demand to entice private sector investments, strengthened public–private partnerships, and deployment of policies that incentivize local investments in R&D [22]. African countries should be included in global vaccine clinical trials on an equal footing, as currently they contribute a paltry 3% of the trials [23].

## 6. Risk Communication and Community Engagement to Combat Vaccine Hesitancy and Misinformation

Additionally, investments in vaccine research and development should be supplemented by community engagement and participation in mpox vaccine deployment. This should be achieved through a well-developed and implemented Risk Communication and Community Engagement (RCCE) strategy. The RCCE strategy should be multidisciplinary and informed by the prevailing epidemiological, contextual, social, economic, cultural, and behavioral considerations. The RCCE strategy could leverage the available traditional and digital media channels [24]. Countries should consider conducting research into social behaviors so as to understand human behaviors and inform the integration of risk communication and behavioral insights into the RCCE strategies [24].

## 7. Strengthen Integrated Surveillance Systems and Conduct Focused Research to Monitor and Track Progress of Herd Immunity

The ability of mpox vaccines to achieve individual and community-level (herd) immunity in various settings needs to be demonstrated and documented. In this regard, studies to assess the efficacy of the candidate mpox vaccines should be supported to provide data to support national rollout of mpox vaccination programs. With successful vaccine rollout, studies to measure the magnitude of any herd effects should become a priority, and this requires a high-quality mpox disease surveillance system. The systems should incorporate new technologies such as pathogen genomics, pathogen genetic sequence data sharing platforms, and advanced technologies like artificial intelligence.

## 8. Financing

The above strategies need to be supported by sustainable financing mechanisms. In the wake of the current global upheaval, securing domestic financial resources to support vaccine development, research, and vaccination campaigns is critical. Establishing vaccination strategies and programs that enhance health security through self-sustaining financial mechanisms and reduced reliance on external donors is essential.

## 9. Conclusions

Approximately 50 years since the smallpox vaccination program was halted following the eradication of smallpox, there has been no dedicated program for sustainable immune boosting. The majority of the world’s young people are at risk, with the Mpox outbreak being a warning sign of broader *Orthopoxvirus* vulnerability. The world urgently needs to restore humanity’s shield against her historically most notorious, indiscriminate, and highly fatal virus family. This is a call to action to rebuild population-level immunity through informed, equitable, and science-based approaches.

## Figures and Tables

**Figure 1 viruses-17-01257-f001:**
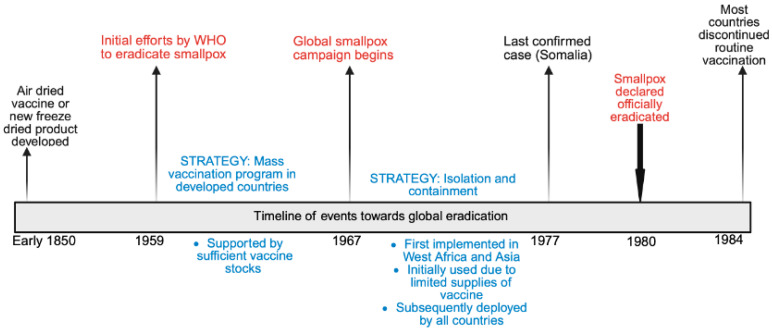
Timeline of events towards global smallpox eradication.

**Table 1 viruses-17-01257-t001:** Summary of studies demonstrating evidence of mpox and cross protection with smallpox and smallpox vaccines.

Title	Authors	Evidence	Sample Size
Evaluation of Cross-Immunity to the Mpox Virus Due to Historic Smallpox Vaccination	Matusali G. et al. [15]	(1) Smallpox vaccinated: Anti-MPXV IgG was detected in 60 individuals (89.6%), while 40 (70.1%) of them had neutralising antibodies. (2) Unvaccinated: Anti-MPXV antibody levels were below the detection limit.	108 (71 smallpox vaccinated, 30 unvaccinated)
Major increase in human monkeypox incidence 30 years after smallpox vaccination campaigns cease in the Democratic Republic of Congo	Rimoin A.W. et al. [8]	(1) Vaccinated persons had a 5.2-fold lower risk of monkeypox than unvaccinated persons (0.78 vs. 4.05 per 10,000). (2) Smallpox vaccine has 80.7% (95% CI: 68.2–88.4) efficacy to prevent mpox and its incidence is inversely correlated with smallpox vaccination.	760 laboratory-confirmed human monkeypox cases
Monkeypox-Induced Immunity and Failure of Childhood Smallpox Vaccination to provide complete protection	Karem L. et al. [16]	Pre-existing immunity, assessed by high anti-Orthopoxvirus IgG levels and childhood smallpox vaccination, was associated (in a nonsignificant manner) with mild disease.	92
Multiple diagnostic techniques identify previously vaccinated individuals with protective immunity against monkeypox	Hammarlund E. et al. [17]	Demonstrated cross-protective antiviral immunity against West African monkeypox can potentially be maintained for decades after smallpox vaccination.	3
Clinical Characteristics of Human Monkeypox, and Risk Factors for Severe Disease	Huhn G.D. et al [18]	Previous smallpox vaccination was not associated with disease severity or hospitalization.	34

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
