# Peer review of "Mpox Epidemics: A Call to Restore Humanity’s Lost Herd Immunity to Orthopoxviruses"

_viruses, 2025, doi:10.3390/v17091257_

Round 1
Reviewer 1 Report
Comments and Suggestions for Authors
The presented manuscript is valuable, but I suggest adding more information/comments about logistical challenges, social barriers, and side effects. Moreover, some chapter titles seem quite long and detailed, so I recommend preparing a shorter form. For example, instead of “Targeted Vaccination with Smallpox Vaccines: Ring Vaccination, Pre-exposure Prophylaxis for High-Risk Groups, and Stockpile Strategies”, a shorter title could be “Targeted Smallpox Vaccination Strategies”. It might be easier to read and understand.
Below, I have outlined my comments:
Line 31 - Poxviridae - Please use italic
Line 36 - Please write out the complete form of the abbreviation WHO.
Figure 1 - The sentence “All countries discontinued routine vaccination” is somewhat imprecise, as not all countries ended routine smallpox vaccination in the same year. For example, France continued until 1984, but most countries in 1980. You might consider revising it.
Please modify Table 1 - the columns are too close together. As a result, the table is difficult to read.
Author Response
Comment: The presented manuscript is valuable, but I suggest adding more information/comments about logistical challenges, social barriers, and side effects. Moreover, some chapter titles seem quite long and detailed, so I recommend preparing a shorter form. For example, instead of “Targeted Vaccination with Smallpox Vaccines: Ring Vaccination, Pre-exposure Prophylaxis for High-Risk Groups, and Stockpile Strategies”, a shorter title could be “Targeted Smallpox Vaccination Strategies”. It might be easier to read and understand.
Response: Thank you for the valuable comments. The suggestion and comment has been addressed.
Comment: Line 31 - Poxviridae - Please use italic
Response: Done
Comment: Line 36 - Please write out the complete form of the abbreviation WHO.
Response: Done
Comment: Figure 1 - The sentence “All countries discontinued routine vaccination” is somewhat imprecise, as not all countries ended routine smallpox vaccination in the same year. For example, France continued until 1984, but most countries in 1980. You might consider revising it.
Response: This has been addressed. “All” has been replaced with “most”.
Comment: Please modify Table 1 - the columns are too close together. As a result, the table is difficult to read.
Response: This has been addressed.
Reviewer 2 Report
Comments and Suggestions for Authors
Misaki Wayengera and his colleagues present a commentary in which they argue for the resumption of vaccinations against orthopoxviruses in light of the recent outbreaks of the orthopox variant MPXV. The text is of particular interest to a wider audience in the context of a special edition on this pathogen. The authors' thoughts, ideas, and demands are well-founded, thoroughly described, and timely. Assuming the authors’ appeal is published, may it also reach the attention of certain governments and societies who are currently postulate the sheer opposite.
A few things need to be changed:
- Strangely enough, the first sentence of the abstract is unreadable. Please rewrite.
- Remove the space within “back-to-back” in line 24
- Skip one of the two “several” in line 32.
- The entire text should be revised with regard to consistent punctuation, especially concerning commas before “and” or “or.”
- About the table: the four columns are misaligned. This issue causes irritating read-throughs such as “HistoricMatusali”, “campainsAnne, or “Fac-Huhn”. “Anne” by the way is the first name of the first author of reference 9, Anne W. Rimoin. Please adjust. This reviewer suggests to fully redo the table and make sure to not extend it over two pages.
- In the references section there are several unnecessary spaces in lines 161, 166, and 190. Remove the green color in line 151. Also, remove the heart-shaped object in line 187. Lloyd-Smith in line158 needs a correction. In line 173, the first author’s initials are TH, not TAEH.
Author Response
Comment: Misaki Wayengera and his colleagues present a commentary in which they argue for the resumption of vaccinations against orthopoxviruses in light of the recent outbreaks of the orthopox variant MPXV. The text is of particular interest to a wider audience in the context of a special edition on this pathogen. The authors' thoughts, ideas, and demands are well-founded, thoroughly described, and timely. Assuming the authors’ appeal is published, may it also reach the attention of certain governments and societies who are currently postulate the sheer opposite.
Response: Thank you for this valuable comment.
Comment: Strangely enough, the first sentence of the abstract is unreadable. Please rewrite.
Response: On our end the sentence reads well: Global efforts to eradicate smallpox—an Orthopoxvirus infection began in the mid-20th century, with the last naturally occurring case reported in 1977.
Comment: Remove the space within “back-to-back” in line 24
Response: This has been done.
Comment: Skip one of the two “several” in line 32.
Response: Thanks. Done
Comment: The entire text should be revised with regard to consistent punctuation, especially concerning commas before “and” or “or.”
Response: This has been rectified throughout the writeup.
Comment: About the table: the four columns are misaligned. This issue causes irritating read-throughs such as “HistoricMatusali”, “campainsAnne, or “Fac-Huhn”. “Anne” by the way is the first name of the first author of reference 9, Anne W. Rimoin. Please adjust. This reviewer suggests to fully redo the table and make sure to not extend it over two pages.
Response: This has been addressed. The table has been readable by adjusting the columns, and the suggestion to sir name for reference 9 has been done.
Comment: In the references section there are several unnecessary spaces in lines 161, 166, and 190.
Response. Done
Comment: Remove the green color in line 151.
Response: Done
Comment: Also, remove the heart-shaped object in line 187.
Response: Done
Comment: Lloyd-Smith in line158 needs a correction.
Response: Done
Comment: In line 173, the first author’s initials are TH, not TAEH.
Response: Done